# Maternal and perinatal outcomes after implementation of a more active management in late- and postterm pregnancies in Sweden: A population-based cohort study

**Karin Källén**[1]*, **Mikael Norman**[2,3], **Charlotte Elvander**[4], **Christina Bergh**[5,6], **Verena Sengpiel**[5,6,7], **Henrik Hagberg**[5,6,7], **Teresia Svanvik**[5,6], **Ulla-Britt Wennerholm**[5,6,7]

1 Institution of Clinical Sciences, Lund, Department of Obstetrics and Gynecology, University of Lund, Lund, Sweden, 2 Division of Pediatrics, Department of Clinical Science, Intervention and Technology, Karolinska Institutet, Stockholm, Sweden, 3 Department of Neonatal Medicine, Karolinska University Hospital, Stockholm, Sweden, 4 Department of Medicine, Unit of Clinical Epidemiology, Karolinska Institute, Stockholm, Sweden, 5 Region Västra Götaland, Sahlgrenska University Hospital, Department of Obstetrics and Gynecology, Gothenburg, Sweden, 6 Institute of Clinical Sciences, Sahlgrenska Academy, University of Gothenburg, Gothenburg, Sweden, 7 Centre of Perinatal Medicine & Health, Sahlgrenska Academy, University of Gothenburg, Gothenburg, Sweden

* karin.kallen@med.lu.se

**Data Availability Statement:** Data cannot be shared publicly because register data are not

## Abstract

### Background

The risk of perinatal death and severe neonatal morbidity increases gradually after 41 weeks of pregnancy. We evaluated maternal and perinatal outcomes after a national shift from expectancy and induction at $42^{+0}$ weeks to a more active management of late-term pregnancies in Sweden offering induction from $41^{+0}$ weeks or an individual plan aiming at birth or active labour no later than $42^{+0}$ weeks.

### Methods and findings

Women with a singleton pregnancy lasting $41^{+0}$ weeks or more with a fetus in cephalic presentation ($N = 150{,}370$) were included in a nationwide, register-based cohort study. Elective cesarean sections were excluded. Outcomes during period 1, January 2017 to December 2019 (before the shift) versus outcomes during period 2, January 2020 to October 1, 2023 (after the shift) were analysed. For comparison, outcomes of pregnancies lasting $39^{+0}$ to $40^{+6}$ weeks ($N = 358{,}548$) were also studied.

Primary outcomes were: First, peri/neonatal death (stillbirth or neonatal death before 28 days); second, composite adverse peri/neonatal outcome (peri/neonatal death, Apgar score <4 at 5 min, hypoxic ischemic encephalopathy grades 1–3, meconium aspiration syndrome, birth trauma, or admission to a neonatal intensive care unit (NICU) ≥4 days); third, composite adverse peri/neonatal outcome excluding admission to NICU; and fourth, emergency

publicly available due to privacy protection, including General Data Protection Regulation (GDPR). Access to Swedish register resources is only granted after ethical review by appropriate authorities. Requests should be directed to registrator@lu.se. Further information on data access for researchers from the Swedish Pregnancy Register data can be found at registerhallare@graviditetsregistret.se, from the Swedish Neonatal Quality Register at dataservice@snq.se, and from Statistics Sweden (https://www.scb.se/vara-tjanster/bestall-dataoch-statistik/). Syntax for the main analyses is shown in Supporting Information (S1 Syntax Main Analysis).

**Funding:** This work has been supported by the Swedish state under the agreement between the Swedish government and the county councils, the ALF-agreement (ALFGBG-70940, CB), (ALFGBG-74830, UBW), (ALFGBG-965174, HH), the Hjalmar Svensson Foundation (UBW), Swedish state under the agreement between the Swedish government and the county councils, the ALF-agreement (ALFGBG-970689, VS), grants from a regional agreement on clinical research (ALF) between Region Stockholm and Karolinska Institutet (RS2022-0674, MN), and the Childhood Foundation of the Swedish Order of Freemasons (MN). The funders had no role in the study design, data collection and analysis, decision to publish, or preparation of the manuscript.

**Competing interests:** MN has received croyalties for Swedish Textbooks in Pediatrics and Neonatology, payments to MN as Medical advisor for the Swedish Patient Insurance (LÖF) and as Associate Editor, Swedish Medical Journal, support to MN for attending meeting/travel from Chiesi Pharma, payment to the clinic for being Director of the Swedish Neonatal Quality Register, stocks in Eva & Mikael Norman AB. CB, HH, UBW, and VS were members of the steering group of the Swedish postterm induction study SWEPIS, one of the trials leading to the changed routine studied in this manuscript. PMID: 31748223. VS is principal coordinating investigator and sponsor representative for the Swedish multicenter OPTION trial - OutPatienT InductiON: Labour induction in an outpatient setting - a multicenter randomized controlled trial. (EU CT no: 2023-507164-39-00, CIV-ID: 20-09-034712). Norgine provides the study drug Angusta. Coloplast labels the Foley balloon catheter used for induction in the OPTION trial as study product, however participating hospitals have to buy the catheters themselves. Neither Norgine or Coloplast had/has any part in writing the study protocol, data collection, analyses or

cesarean section. Secondary outcomes included the components of the primary composite outcomes. Relative risks (RRs) with 95% confidence intervals (CIs) for binary outcomes period 2 versus period 1 were computed using modified Poisson regression analyses with adjustments for maternal age, parity, body mass index (BMI), smoking, and educational level.

Induction rates among pregnancies lasting $41^{+0}$ weeks or more increased from 33.7% in period 1 to 52.4% in period 2. Mean (standard deviation) gestational age at birth decreased from 290.7 (2.9) days to 289.6 (2.3) days. Infants born during period 2 were at lower risk of peri/neonatal death compared to infants born during period 1; 0.9/1,000 versus 1.7/1,000 born infants (adjusted RR 0.52; 95% CI [0.38, 0.69]; $p < 0.001$), and they had a lower risk of having the composite adverse neonatal outcome, both including (50.5/1,000 versus 53.9/1,000, adjusted RR 0.92; 95% CI [0.88, 0.96]; $p < 0.001$) or excluding NICU admission (18.5/1,000 versus 22.5/1,000, adjusted RR 0.79; 95% CI [0.74, 0.85]; $p < 0.001$). The cesarean section rate increased from 10.5% in period 1 to 11.9% in period 2 (adjusted RR 1.07; 95% CI [1.04, 1.10]; $p < 0.001$). For births at 39 to 40 weeks the adjusted RR for peri/neonatal death was 0.86 (95% CI [0.72, 1.02]). One limitation of the study is that we had no data on to what extent monitoring of fetal health was performed.

## Conclusions

A more active management of pregnancies lasting $41^{+0}$ weeks or more was associated with a decrease in peri/neonatal deaths, and a decrease in composite adverse peri/neonatal outcomes. Increased rate of emergency cesarean sections was observed. Women with pregnancies advancing towards 41 gestational weeks should be given balanced information on the benefits and risks of induction of labour at 41 weeks compared to expectant management until 42 weeks and be offered induction of labour at 41 weeks or active surveillance of pregnancies from 41 weeks in order to decrease peri/neonatal mortality.

## Author summary

### Why was this study done?

- Rare but severe outcomes such as stillbirth, perinatal mortality, and severe neonatal and maternal morbidity are more common among late- and postterm pregnancies than among full-term pregnancies.

- Randomised clinical trials have shown decreased perinatal mortality by labour induction of late- and postterm pregnancies while real-world observational studies have shown conflicting results after such an intervention.

### What did the researchers do and find?

- We have performed a register-based national study, which confirms earlier randomised clinical trials, demonstrating decreased rates of peri/neonatal mortality and a composite outcome of peri/neonatal mortality and severe neonatal morbidity including birth

publication of results. KK, CE and TS have declared no conflicts of interest.

**Abbreviations:** BMI, body mass index; CI, confidence interval; CTG, cardiotocography; HIE, hypoxic ischaemic encephalopathy; NICU, neonatal intensive care unit; RR, relative risk; SPR, Swedish Pregnancy Register; SWEPIS, Swedish Postterm Induction Study; WHO, World Health Organization.

trauma after offering pregnant women induction of labour or an individual plan from 41 instead of 42 gestational weeks.

- An increase in cesarean sections was also observed.

## What do these findings mean?

- Women with pregnancies advancing towards 41 gestational weeks should be given balanced information on the benefits and risks of induction of labour at 41 weeks compared to expectant management until 42 weeks and be offered induction of labour at 41 weeks or active surveillance of pregnancies from 41 weeks in order to decrease peri/neonatal mortality.

- One limitation of the study is that we had no data on to what extent monitoring of fetal health was performed.

## Introduction

Obstetric management of prolonged pregnancy is a challenge. Rare but severe outcomes such as stillbirth, perinatal mortality, and severe neonatal and maternal morbidity are more common among prolonged pregnancies than among full-term pregnancies (between 39 weeks+0 days ($39^{+0}$ weeks) and 40 weeks+6 days ($40^{+6}$ weeks)) and increases as gestational age advances [1–6]. Induction of labour (induction) is used to reduce these adverse effects as shown in the ARRIVE trial [7] but is controversial since in observational studies the procedure, when compared to spontaneous onset of labour, has been associated with harm such as increased need for interventions during labour, uterine rupture, and adverse neonatal outcome [8,9].

Systematic reviews and randomised clinical trials suggest that induction before 42 gestational weeks may reduce perinatal mortality and severe neonatal morbidity without increasing the frequency of cesarean section, operative vaginal delivery, or postpartum haemorrhage [10–15]. Furthermore, the ARRIVE trial showed that induction at 39 weeks in low-risk nulliparous women resulted in a significantly lower frequency of cesarean section but did not change the composite adverse perinatal outcome [7].

The Swedish randomised controlled trial (Swedish Postterm Induction Study (SWEPIS)) included 2,760 women with a low-risk singleton pregnancy before it was stopped before completion because of significantly higher perinatal mortality in the expectant management group than in the early induction group [15]. Data from observational studies of a change in strategy to a more active induction policy in postdate pregnancies have shown conflicting results in relation to stillbirth and perinatal mortality [9,16]. Furthermore, reliable information is still lacking if maternal characteristics such as age, parity, and BMI moderate the effect of induction strategies on adverse outcomes in postdate pregnancies [5,11,17].

Several current international clinical practice guidelines recommend induction between $41^{+0}$ weeks and $42^{+0}$ weeks [18–21]. The World Health Organization (WHO) [22] recommends induction at $41^{+0}$ weeks, whereas the Norwegian guidelines published in 2024 recommend a consultation at $41^{+0}$ to $41^{+3}$ weeks and labour induction at $42^{+0}$ weeks at the latest [23]. The publication of SWEPIS in 2019 started a debate in Sweden about management of late-term pregnancies finally summarised in national guidelines in 2021, which recommended

that all women should be offered either induction from 41 completed weeks or an individual plan aiming at birth or active labour latest at 42 completed weeks [24].

Labour induction is an intervention that can potentially cause both benefit and harm; thus, it is important to investigate the effects of this new management on a national level. The aim of this study was to evaluate the maternal and neonatal outcomes after a national change in management of late-term pregnancies in Sweden. The hypothesis was that early induction improved perinatal outcome. Furthermore, we wanted to assess if the outcomes differ according to hospital practice, i.e., low or high rates of change in labour inductions as well as if outcomes differ depending on the individual characteristics of the women.

## Methods

This study is reported as per the Reporting of Studies Conducted using Observational Routinely Collected Data (RECORD) (**S1 RECORD Checklist**).

### Data sources

Data with information on pregnancy as well as pregnancy and neonatal outcomes were obtained from the Swedish Pregnancy Register (SPR) [25] and the Swedish Neonatal Quality Register (SNQ) [26]. Information on mortality was retrieved from Statistics Sweden. Linkages were made using the unique personal identification number designated to everyone with residence permit in Sweden.

SPR is a nationwide quality register with data from almost all maternal health care units and delivery wards in Sweden, including more than 95% of all deliveries in Sweden [25]. Data are compiled from electronic standardised prenatal, delivery, and birth records including neonatal diagnoses. At the first antenatal visit (usually at 8 to 12 gestational weeks), each woman is interviewed by a midwife, and information on, e.g., height, weight, BMI, smoking, country of birth, chronic maternal diseases, and educational level is registered. Data on pregnancy and neonatal outcomes are registered at each delivery hospital and are electronically reported to SPR. Women rate their overall childbirth experience on a 10-point numeric rating validated scale where childbirth experience is defined as negative (ratings 1–4), mixed (ratings 5–6), or positive (ratings 7–10) [27,28]. Self-rated health is evaluated with a one-item question where the woman rates her health status as very good, good, neither good nor bad, bad, very bad, which has been shown to be a useful marker of objectively measured health status. Poor self-rated health is defined as ratings of "very poor," "poor," or "neither good nor poor," whereas good self-reported health is defined as ratings of "good" or "very good" [25,29].

SNQ is a national quality register that contains data on all admissions for neonatal care in Sweden [26]. Individual patient data are electronically reported daily, and data include information on neonatal conditions, diagnoses, and medical and surgical interventions. In both SPR and SNQ, diagnoses are reported as International Classification of Diseases 10th revision (ICD-10)–codes.

### Study population

Singleton pregnancies lasting $39^{+0}$ gestational weeks or more, according to ultrasound-based dating in the first or early second trimester or for pregnancies after assisted reproduction after the day of embryo transfer, with fetuses in cephalic presentation at birth, and births registered in SPR between January 1, 2017 and October 1, 2023 were included. Elective cesarean sections were excluded. Pregnancies lasting $41^{+0}$ weeks or more constituted the study group. Information on pregnancies lasting $39^{+0}$ to $40^{+6}$ weeks were analysed for comparison.

## Study design

The overall purpose was to evaluate the pregnancy and neonatal outcomes in the study population before and after introduction of a more active management to women who reached pregnancy week $41^{+0}$ to $41^{+2}$ offering either induction of labour or an individual plan aiming at birth or active labour latest at 42 completed weeks. The main analyses compared outcomes during the period before (period 1; January 1, 2017 to December 31, 2019) with the period after the new strategy was introduced (period 2; January 1, 2020 to October 1, 2023).

## Outcomes

**Primary outcomes.**   First, peri/neonatal death (stillbirth or neonatal death before 28 days); second, composite adverse peri/neonatal outcome (peri/neonatal death, Apgar score <4 at 5 min [30], hypoxic ischaemic encephalopathy [HIE] grades 1–3, meconium aspiration syndrome, birth trauma [ICD-10: P10 –P15], or admission to a neonatal unit for 4 days or more [31], thus excluding neonates admitted only for surveillance at a neonatal unit); third, composite adverse peri/neonatal outcome excluding neonatal admission; and fourth, emergency cesarean section.

**Secondary outcomes.**   All subcomponents defining the primary outcomes with additions: Apgar score <7 at 5 min, HIE grades 2 and 3, instrumental birth (vacuum extraction or forceps), obstetric anal sphincter rupture, defined as a third- or fourth-degree perineal injury involving the external or internal anal sphincter muscles, or both, as defined by the codes O702 and O703 in the Swedish version of the International Classification of Diseases 10th edition, postpartum haemorrhage >1,000 ml, (as reported in ml), and endometritis. Additional secondary outcome measures were the maternal experience of childbirth measured before discharge from hospital a few days after birth and self-rated health 2 months after delivery, respectively.

## Statistical analysis

The prospective study protocol detailing the planned analysis can be found in **S1 Protocol**. Changes from the planned protocol as response to peer review comments were (1) multiple imputation used to replace missing information on BMI or smoking for the main outcomes; and (2) the additional analysis to control for possible difference in hospital practice. Syntax for the main analyses is shown in **S1 Syntax main analysis**.

Chi-square tests were used to evaluate heterogeneity between groups displayed in descriptive tables. Relative risks (RRs) for binary outcomes period 2 versus period 1 were computed using modified Poisson regression analyses. To control for possible difference in hospital practice, additional analyses were carried out where the results before and after the breaking point were stratified by the k units, and overall RRs were computed in which the unit-specific log (RRs) were weighted by precision (1/Standard Error of log ($RR_k$)). Adjustments were made for maternal age (continuous), parity (primiparity, multiparity without previous cesarean section, previous cesarean section), BMI (continuous), educational level (class variables: elementary school or less, 3 years post elementary school, university, or not known), and maternal smoking during pregnancy (ordinal, treated as continuous: 1 = no, 2 = 1–9 cigarettes per day, 3 = 10 or more cigarettes per day). In the main analyses (primary outcomes), multiple imputation was used to replace missing information on BMI or smoking. In the other analyses, missing information on BMI and smoking was replaced by the overall mean. Tests of homogeneity of adjusted $RR_i$s over k strata (e.g., over parity-strata or BMI-para strata) were performed where the log (RRs) were weighted by precision (1/Standard Error of log ($RR_k$)) and compared to the Chi2(k-1) distribution. To detect a possible linear trend of log ($RR_i$)s along an independent

ordinal variable (like hospital change-tertile strata or BMI-class), a weighted linear regression of the log (RR$_i$s) was carried out, using the same weights as previously specified (1/Standard error of log (RR$_i$)).

Statistical analyses were performed using IBM SPSS Statistics, version 27.0. (Armonk, NY: IBM Corp, United States of America) and Gauss (Aptech Systems Inc., Arizona, USA).

## Sensitivity analyses

First, sensitivity analyses were performed evaluating the outcomes where hospital-specific breaking points for changed management were used instead of the fixed periods (2017–2019 and 2020–2023, respectively). These analyses were initiated because adaptions to the new recommendations were not simultaneous among the Swedish hospitals. The breaking points were defined as the first quarter for which the hospital-specific induction rate within the next 6 months was at least 30% above the hospital-specific induction rate during baseline (2017–2018).

A second set of sensitivity analyses evaluated the impact of the year-and-hospital-specific induction rate among pregnancies $41^{+0}$ weeks or more on outcomes. In these analyses, the actual induction rate per year (during period 1 and period 2) and hospital was recorded and sorted into tertiles, where tertile 1 included deliveries at hospitals and years with the lowest induction rates, and tertile 3 included deliveries at hospitals and years with the highest induction rates.

Third, the Swedish delivery hospitals were categorised into 3 tertiles according to the magnitude of the induction rate change among pregnancies $41^{+0}$ weeks or more between the time periods and the relative risks (period 2 versus period 1) for outcomes were individually computed for each hospital tertile category.

Fourth, we investigated if the magnitude of the association between time period and outcomes was homogenous over maternal BMI, parity, and country-of-birth strata.

The fifth sensitivity was performed to explore if the results were mainly influenced by the result among pregnancies lasting $42^{+0}$ weeks or more, by stratifying by gestational duration, births $41^{+0}$ to $41^{+6}$ and $42^{+0}$ weeks or more, respectively.

In the sixth set of sensitivity analyses, the RRs (period 2 versus period 1) for selected outcomes among pregnancies lasting $41^{+0}$ weeks or more were compared with the corresponding RRs among pregnancies lasting $39^{+0}$ to $40^{+6}$ weeks.

Seventh, in order to exclude any interference in outcomes from a revised intrapartum cardiotocography (CTG) classification introduced in 2018 [32], one sensitivity analysis also restricted period 1 to 2018–2019.

In the final sensitivity analysis, infants with significant congenital malformations were excluded [33].

## Ethics

This study was approved by the Swedish Ethical Review Authority (Dnr 2023-04274-01). This authority also waived patient informed consent. All participants have an opt-out option in the SPR and SNQ.

## Patient involvement

No patients were involved in defining the research question or outcome measures, nor were they asked to give advice on interpretation of results.

### Dissemination to participants and related patient and public communities

This is a register-based study based on pseudonymised data and no feedback will be given to individual patients on the results of the study. Dissemination to the Swedish population (which constitutes the study population) will be through media outreach (e.g., press release and communication), presentations at national and international forums, and reports to the Swedish Pregnancy Register, the Swedish Neonatal Quality Register, the National system for knowledge-driven management within Swedish healthcare, and academic societies on publication of this study.

## Results

In total, there were 543,744 deliveries after $39^{+0}$ weeks of pregnancy or more. Multiple births, non-cephalic presentations, and elective cesarean sections were excluded. The cohort selection process is summarised in **Fig 1**. Among pregnancies lasting at least $39^{+0}$ weeks, the rate of deliveries at $41^{+0}$ weeks or more decreased from 31% during 2017 to 2019 to 28% during 2020 to 2023 ($p$-value for difference $<0.001$).

**Table 1** shows the maternal and pregnancy characteristics by period of delivery (2017–2019 or 2020–2023) among pregnancies lasting $41^{+0}$ weeks or more. The differences in characteristics by period of delivery were small in absolute measures. The induction rate in this cohort increased from 33.7% to 52.4% from period 1 to period 2. Correspondingly, the rate of pregnancies lasting $42^{+0}$ weeks or more in this population decreased from 24.0% to 7.7%.

**Fig 2** (left) shows the induction rate by time period and gestational week (3 days interval) on the x-axis. A very prominent increase of the induction rate between the 2 time periods (2017–2019 and 2020–2023, respectively) was noted among births at $41^{+1}$ to $41^{+6}$ weeks (from 21% to 55%, $p < 0.001$). Among deliveries at $39^{+0}$ to $41^{+0}$ weeks, the induction rate was significantly higher during period 2 than period 1 (from 14% to 20%, $p < 0.001$), but among deliveries at $42^{+0}$ weeks or more, no major change in induction rates between the 2 study periods was observed (from 77% to 82%, $p < 0.001$).

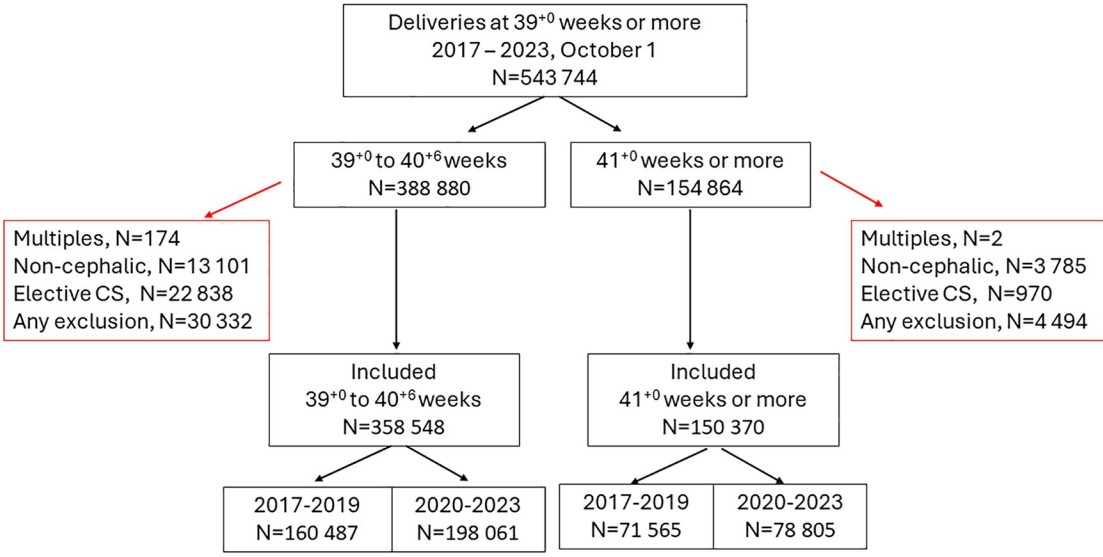

**Fig 1. Flowchart showing the cohort selection process.**

**Table 1. Maternal and pregnancy characteristics by period of delivery among pregnancies lasting 41$^{+0}$ weeks or more.** Included were singleton pregnancies in cephalic presentation at birth. Elective cesarean sections were excluded.

| | | 2017–2019 N = 71,565 | | 2020–2023* N = 78,805 | |
|---|---|---|---|---|---|
| | | *n* | (%) | *n* | (%) |
| Labour induction | | 24,123 | (33.7) | 41,297 | (52.4) |
| Postterm deliveries (≥42$^{+0}$ weeks) | | 17,202 | (24.0) | 6,034 | (7.7) |
| Age (years) | | | | | |
| | <25 | 7,907 | (11.0) | 6,681 | (8.5) |
| | 25–34 | 48,721 | (68.1) | 55,165 | (70.0) |
| | 35–39 | 10,904 | (15.2) | 12,706 | (16.1) |
| | 40+ | 2,546 | (3.6) | 2,713 | (3.4) |
| Reproductive history | | | | | |
| | Primipara | 35,152 | (49.1) | 40,484 | (51.4) |
| | Multipara, no CS | 31,482 | (44.0) | 33,123 | (42.0) |
| | Previous CS | 4,931 | (6.9) | 5,198 | (6.6) |
| BMI$^{†}$ | | | | | |
| | <25 | 37,285 | (52.1) | 41,255 | (52.4) |
| | 25–29 | 18,753 | (26.2) | 22,479 | (28.5) |
| | 30–34 | 7,156 | (10.0) | 8,444 | (10.7) |
| | 35+ | 3,229 | (4.5) | 3,751 | (4.8) |
| | Not known | 5,142 | (7.2) | 2,876 | (3.6) |
| Smoking | | | | | |
| | Nonsmoking | 64,072 | (89.5) | 72,995 | (92.6) |
| | Smoking | 2,169 | (3.0) | 1,920 | (2.4) |
| | Smoking not known | 5,324 | (7.4) | 3,890 | (4.9) |
| Country of birth | | | | | |
| | Nordic country$^{‡}$ | 48,410 | (67.6) | 53,440 | (67.8) |
| | Europe outside Nordic countries | 4,514 | (6.3) | 5,082 | (6.4) |
| | Outside Europe | 13,238 | (18.5) | 14,318 | (18.2) |
| | Not known | 5,403 | (7.5) | 5,965 | (7.6) |
| Educational level | | | | | |
| | Compulsory school 9 years or less | 5,329 | (7.4) | 4,620 | (5.9) |
| | Secondary school, 12 years of education | 22,413 | (31.3) | 22,049 | (28.0) |
| | University | 33,439 | (46.7) | 40,056 | (50.8) |
| Assisted reproduction | | 2,610 | (3.6) | 3,410 | (4.3) |
| Pregnancy complications | | | | | |
| | Pregestational diabetes | 136 | (0.2) | 153 | (0.2) |
| | Gestational diabetes | 856 | (1.2) | 2,331 | (3.0) |
| | Gestational hypertension/preeclampsia | 1,472 | (2.1) | 2,143 | (2.7) |
| Child characteristics | | | | | |
| | Sex, male | 39,097 | (54.6) | 42,630 | (54.1) |
| | Birth weight >4,500 g | 4,757 | (6.6) | 4,459 | (5.7) |

* 2020-01-01 to 2023-10-01.

$^{†}$BMI = kg/m$^2$; kg is a person's weight in kilograms and m$^2$ is height in metres squared.

$^{‡}$Nordic countries include Denmark, Finland, Iceland, Norway, and Sweden.

BMI, body mass index; CS, cesarean section.

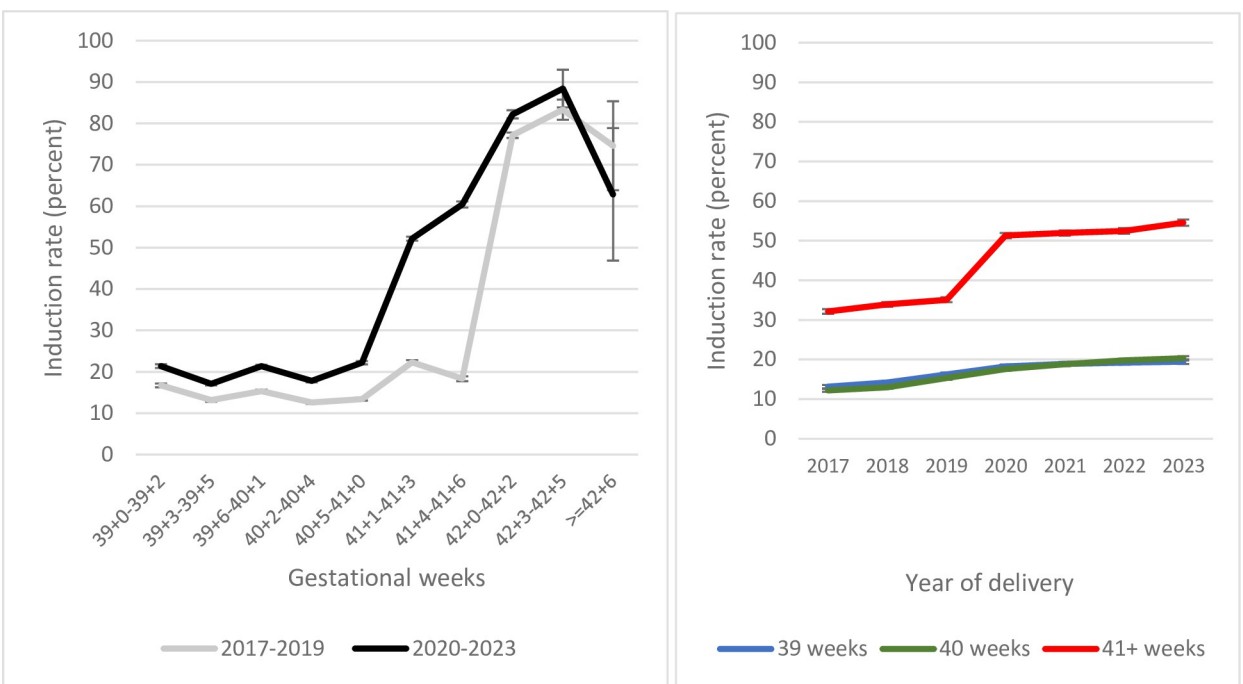

**Fig 2.** Induction rate by gestational duration (left) and year of birth (right) on the x-axis (respectively) among term singleton pregnancies lasting $39^{+0}$ weeks or more, cephalic presentations at birth. Elective cesarean sections were excluded. Sweden 2017 to October 1, 2023. The vertical bars represent 95% CIs.

Fig 2 (right) shows the induction rate by gestational duration and year of delivery on the x-axis. For births between 2017 and 2023, the induction rate increased gradually among deliveries at 39 or 40 completed weeks of pregnancies, from 13% in 2017 to 20% in 2023. For deliveries $\geq 41^{+0}$ weeks, a rapid change indicating the implementation of the new active management of late-term pregnancies was observed between 2019 and 2020, from 35% to 51%. Among pregnancies lasting $41^{+0}$ or more, the mean (standard deviation) gestational age at birth decreased marginally, from 290.7 (2.9) days in period 1 to 289.6 (2.3) in period 2 (difference 1.13 days, 95% CI [1.10, 1.16]).

There were, however, variations among the hospitals regarding the timing and the implementation of the new recommendations.

The magnitude of the induction rate increase between period 1 and 2 was similar among nulliparous and parous women giving birth at $41^{+0}$ weeks or more: Among nulliparous women, the induction rate increased from 38.7% in period 1 to 58.7% in period 2 (RR 1.50; 95% CI [1.47, 1.54]). The corresponding figures for parous women (women with previous cesarean section excluded) were 27.5% and 45% (RR 1.64; 95% CI [1.59, 1.69]).

Table 2 shows the neonatal outcomes, organised in primary and secondary outcomes, numbers and risk ratio, delivery period 2 (2020 to 2023) versus period 1 (2017 to 2019). Adjustments for age, parity, previous cesarean section, BMI, educational level, and smoking changed the obtained RRs only marginally. Compared to infants born during 2017 to 2019, infants born during 2020 to 2023 were at significantly lower risk of peri/neonatal death (adjusted RR 0.52; 95% CI [0.38, 0.69]; $p < 0.001$), stillbirth (adjusted RR 0.46; 95% CI [0.33, 0.64]; $p < 0.001$), perinatal death (adjusted RR 0.54; 95% CI [0.40, 0.72]; $p < 0.001$), and to suffer from birth trauma (adjusted RR 0.80; 95% CI [0.73, 0.87]; $p < 0.001$). Adjustments for cesarean section or birth weight did not have any major impact on the RR for birth trauma,

**Table 2. Neonatal outcomes.** Significant findings are indicated in bold text.

| | Delivery period | | | | Risk ratio (RR) period 2 versus period 1 | | | |
| | 2017–2019 N = 71,565 | | 2020–2023* N = 78,805 | | Crude | | Adjusted[†] | |
| | *n* | Per thousand | *n* | Per thousand | RR | 95% CI; *p*-value | RR | 95% CI; *p*-value |
|---|---|---|---|---|---|---|---|---|
| Primary outcomes | | | | | | | | |
| Peri/neonatal death | 124 | (1.7) | 74 | (0.9) | **0.54** | **0.41, 0.72; *p* < 0.001** | **0.52** | **0.38, 0.69; *p* < 0.001** |
| Composite outcome[‡] | 3,856 | (53.9) | 3,978 | (50.5) | **0.94** | **0.90, 0.98; *p* < 0.001** | **0.92** | **0.88, 0.96; *p* < 0.001** |
| Composite II[§] | 1,609 | (22.5) | 1,460 | (18.5) | **0.82** | **0.77, 0.88; *p* < 0.001** | **0.79** | **0.74, 0.85; *p* < 0.001** |
| Secondary outcomes | | | | | | | | |
| Stillbirth | 101 | (1.4) | 52 | (0.7) | **0.47** | **0.34, 0.65; *p* < 0.001** | **0.46** | **0.33, 0.64; *p* < 0.001** |
| Perinatal death | 120 | (1.7) | 73 | (0.9) | **0.55** | **0.41, 0.74; *p* < 0.001** | **0.54** | **0.40, 0.72; *p* < 0.001** |
| Neonatal death[¶] | 23 | (0.3) | 22 | (0.3) | 0.87 | 0.48, 1.56; *p* = 0.635 | 0.86 | 0.48, 1.53; *p* = 0.605 |
| Apgar 5 min <4[¶] | 289 | (4.0) | 288 | (3.7) | 1.00 | 0.84, 1.20; *p* = 0.963 | 0.98 | 0.82, 1.18; *p* = 0.858 |
| Apgar 5 min <7[¶] | 1,139 | (15.9) | 1,293 | (16.4) | 1.06 | 0.98, 1.15; *p* = 0.168 | 1.04 | 0.96, 1.12; *p* = 0.390 |
| NICU ≥ 4 days[¶] | 2,759 | (38.6) | 2,995 | (38.0) | 0.98 | 0.94, 1.04; *p* = 0.561 | 0.97 | 0.92, 1.02; *p* = 0.207 |
| Meconium aspiration[¶] | 185 | (2.6) | 175 | (2.2) | 0.86 | 0.70, 1.06; *p* = 0.147 | 0.83 | 0.68, 1.02; *p* = 0.085 |
| Birth trauma[¶] | 1,065 | (14.9) | 961 | (12.2) | **0.82** | **0.75, 0.89; *p* < 0.001** | **0.80** | **0.73, 0.87; *p* < 0.001** |
| HIE1-3[¶] | 196 | (2.7) | 196 | (2.5) | 0.91 | 0.74, 1.11; *p* = 0.336 | 0.87 | 0.71, 1.06; *p* = 0.164 |
| HIE2-3[¶] | 93 | (1.3) | 85 | (1.1) | 0.83 | 0.62, 1.11; *p* = 0.212 | 0.80 | 0.60, 1.08; *p* = 0.146 |

*2020-01-01 to 2023-10-01.

[†]Adjusted for maternal age, parity, previous cesarean section, body mass index, educational level, and smoking.

[‡]Composite outcome: Peri/neonatal death, Apgar 5 min 0–3, NICU ≥4 days, HIE 1–3, Meconium aspiration, or delivery trauma.

[§]Composite II: Like Composite, but without criteria NICU ≥4 days.

[¶]Stillbirths excluded from analysis.

CI, confidence interval; HIE, hypoxic ischaemic encephalopathy; NICU, neonatal intensive care unit.

period 2 versus period 1. Infants born during period 2 were also at significantly lower risk to have an adverse composite outcome as defined in the Methods (adjusted RR 0.92; 95% CI [0.88, 0.96]; *p* < 0.001, and adjusted RR 0.79; 95% CI [0.74, 0.85]; *p* < 0.001 for the composite outcome including or not including neonatal care, respectively). For the primary outcomes, additional analyses were carried out considering the unit-specific before and after results. The results from these analyses are shown in **S1 Table** and are almost identical to the results displayed in **Table 2**.

There were no statistical differences in the risks of low Apgar score, admission to neonatal care ≥4 days, meconium aspiration, or HIE between the 2 study periods.

**Table 3** shows the maternal outcomes by numbers and RR, delivery period 2 versus period 1. There was a small, albeit statistically significant increase of emergency cesarean section, also after adjustments for possible confounders (adjusted RR 1.07; 95% CI [1.04, 1.10]; *p* < 0.001). The crude RR for vacuum extraction/forceps was significantly above unity, but the adjusted estimate did not reach statistical significance. Among women who delivered vaginally, the risk of obstetric anal sphincter rupture was significantly lower during period 2 than period 1 (adjusted RR 0.90; 95% CI [0.85 to 0.95]; *p* < 0.001). The risk of postpartum haemorrhage >1,000 ml increased significantly (adjusted RR 1.07; 95% CI [1.04, 1.10]; *p* < 0.001), although the change was minor in absolute measures. In addition, adjustment for cesarean section did not change this estimate (adjusted RR 1.06; 95% CI [1.03 to 1.09]). The corresponding adjusted RR for postpartum haemorrhage >1,000 ml among deliveries at $39^{+0}$ to $40^{+6}$ weeks was 1.10 (95% CI [1.07, 1.12]). The rate of endometritis did not change between the time periods.

**Table 3. Delivery mode and maternal outcomes.** Significant findings are indicated in bold text.

| | | Delivery period | | | | Risk ratio (RR) period 2 versus period 1 | | | |
| --- | --- | --- | --- | --- | --- | --- | --- | --- | --- |
| | | 2017–2019 N = 71,565 | | 2020–2023* N = 78,805 | | Crude | | Adjusted[†] | |
| | | n | (%) | n | (%) | RR | 95% CI; p-value | RR | 95% CI; p-value |
| Primary outcome | | | | | | | | | |
| | Emergency CS | 7,550 | (10.5) | 9,367 | (11.9) | **1.13** | **1.10, 1.16; p < 0.001** | **1.07** | **1.04, 1.10; p < 0.001** |
| Secondary outcomes | | | | | | | | | |
| | VE/Forceps | 5,617 | (7.8) | 6,566 | (8.3) | **1.06** | **1.03, 1.10; p = 0.001** | 1.02 | 0.98, 1.05; p = 0.361 |
| | Vaginal non-instrumental delivery | 58,398 | (81.6) | 62,872 | (79.8) | **0.98** | **0.97, 0.98; p < 0.001** | **0.99** | **0.98, 0.99; p < 0.001** |
| | Sphincter rupture[‡] | 2,387 | (3.7) | 2,401 | (3.5) | **0.93** | **0.88, 0.98; p = 0.008** | **0.90** | **0.85, 0.95; p < 0.001** |
| | Endometritis | 699 | (1.0) | 731 | (0.9) | 0.95 | 0.86, 1.05; p = 0.327 | 0.94 | 0.85, 1.04: p = 0.228 |
| | Haemorrhage >1,000 ml | 6,564 | (9.2) | 7,947 | (10.1) | **1.10** | **1.07, 1.13; p < 0.001** | **1.07** | **1.04, 1.10; p < 0.001** |
| | Negative childbirth experience | 3,156 | (7.3)[§] | 3,689 | (6.5)[¶] | **0.88** | **0.85, 0.93; p < 0.001** | **0.87** | **0.83, 0.91; p < 0.001** |
| | Poor self-rated health after birth | 4,044 | (7.5)[**] | 4,677 | (7.9)[††] | **1.04** | **1.00, 1.08; p = 0.047** | 1.02 | 0.98, 1.06; p = 0.274 |

*2020-01-01 to 2023-10-01.

[†]Adjusted for maternal age, parity, previous CS, BMI, educational level, and smoking.

[‡]CS excluded from analyses.

[§]N = 42,682 (response rate = 60%).

[¶]N = 56,344 (response rate = 71%).

[**]N = 48,317 (response rate = 68%).

[††]N = 53,644 (response rate = 68%).

BMI, body mass index; CI, confidence interval; CS, cesarean section; VE, vacuum extraction.

In this cohort of women giving birth at $41^{+0}$ weeks or more, the proportion of women who reported a negative childbirth experience was slightly lower among women giving birth in period 2 compared to period 1 (6.5% as compared to 7.5% in the first period, adjusted RR 0.87; 95% CI [0.83, 0.91]; $p < 0.001$). The proportion of women who reported a poor self-rated health after childbirth (7.9% versus 7.5%) did not change between the periods (adjusted RR 1.02; 95% CI [0.98, 1.06]; $p = 0.274$).

The adjusted RR of 0.52 for peri/neonatal death, period 2 versus period 1, corresponds to an extra survival of approximately 8 children per 10,000 births. Similarly, the adjusted RR of 1.07 for emergency cesarean section corresponds to an extra 74 cesarean sections per 10,000 births. And, finally, the increased induction rate from 33.3% to 52.4% corresponds to approximately 1,900 more inductions per 10,000 deliveries at $41^{+0}$ weeks or more.

## Sensitivity analyses

Using the detailed hospital specific breaking points for implementation of the new induction policy, the adjusted RR for peri/neonatal death was 0.52 (95% CI [0.38, 0.71]), thus almost identical to the corresponding adjusted RR for period 2 versus period 1. Infants who were born in hospitals belonging to the upper induction-rate tertile (mean induction rate 62%, range 45% to 91%) were at significantly lower risk of peri/neonatal death compared to infants born at hospitals belonging to the lowest induction-rate tertile (mean induction rate 28%, range 11% to 32%, adjusted RR 0.50 (95% CI [0.33, 0.75]).

Table 4 shows the results from sensitivity analyses, estimating the RR for peri/neonatal death and emergency cesarean section, respectively, stratified by different groups (hospital induction change tertiles, parity, BMI, country of birth, gestational weeks). The table shows

**Table 4. Result from sensitivity analyses.** Adjusted risk ratios (ARR) by strata as specified. Significant findings are indicated in bold text.

| | Strata | ARR* for peri/neonatal death, period 2 versus period 1 | | p-Value[†] | ARR* for emergency CS, period 2 versus period 1 | | p-Value[†] |
|---|---|---|---|---|---|---|---|
| | | ARR | 95% CI | | ARR | 95% CI | |
| **Deliveries 41[+0] weeks or more** | | | | | | | |
| Hospital induction change tertile[‡] | | | | 0.101[§] | | | 0.445[§] |
| | Tertile least change | 0.64 | 0.41, 0.98 | | 1.06 | 1.02, 1.10 | |
| | Middle tertile | 0.56 | 0.36, 0.89 | | 1.10 | 1.05, 1.15 | |
| | Tertile largest change | 0.29 | 0.14, 0.63 | | 1.06 | 1.00, 1.13 | |
| Reproductive history | | | | 0.638 | | | **0.047** |
| | Primipara | 0.47 | 0.32, 0.69 | | 1.06 | 1.03, 1.10 | |
| | Multipara without CS | 0.60 | 0.35, 1.01 | | 1.03 | 0.94, 1.13 | |
| | Previous CS | 0.65 | 0.32, 1.36 | | 1.15 | 1.08, 1.22 | |
| BMI[¶] | | | | 0.156[§] | | | 0.158[§] |
| | <25 | 0.62 | 0.40, 0.98 | | 1.09 | 1.04, 1.06 | |
| | 25–29.9 | 0.52 | 0.31, 0.89 | | 1.02 | 0.97, 1.07 | |
| | > = 30 | 0.36 | 0.20, 0.74 | | 1.07 | 1.01, 1.14 | |
| Para-BMI-class | | | | 0.377 | | | 0.221 |
| | Primipara, BMI <30 | 0.57 | 0.36, 0.89 | | 1.07 | 1.03, 1.10 | |
| | Primipara, BMI 30+ | 0.26 | 0.11, 0.61 | | 1.06 | 0.98, 1.14 | |
| | Multipara, BMI <30 | 0.71 | 0.38, 1.33 | | 1.04 | 0.93, 1.15 | |
| | Multipara, BMI 30+ | 0.40 | 0.15, 1.04 | | 0.98 | 0.81, 1.17 | |
| | Previous CS | 0.66 | 0.32, 1.36 | | 1.15 | 1.08, 1.22 | |
| Gestational weeks | | | | 0.823 | | | 0.064 |
| | 41[+0] to 41[+6] | 0.53 | 0.39, 0.72 | | 1.21 | 1.17, 1.25 | |
| | 42[+0] or more | 0.59 | 0.25, 1.40 | | 1.13 | 1.06, 1.20 | |
| Maternal country of birth | | | | 0.770 | | | 0.188 |
| | Nordic country** | 0.54 | 0.37, 0.79 | | 1.06 | 1.02, 1.10 | |
| | Europe outside Nordic countries | 0.86 | 0.28, 2.68 | | 1.06 | 0.97, 1.16 | |
| | Outside Europe | 0.49 | 0.28, 0.86 | | 1.05 | 0.93, 1.18 | |
| | Not known | 0.40 | 0.15, 1.03 | | 1.05 | 0.95, 1.16 | |
| **All deliveries 39+0 weeks or more** | | | | **0.004** | | | **0.006** |
| | Week 39[+0]–40[+6] | 0.86 | 0.72, 1.02 | | 1.14 | 1.11–1.18 | |
| | Week ≥41[+0] | 0.53 | 0.40, 0.71 | | 1.08 | 1.05–1.11 | |

*Adjusted for maternal age, parity, previous cesarean section, BMI, educational level, and smoking

[†]p-Value for homogeneity among risk ratios if not otherwise specified.

[‡]The Swedish delivery hospitals were categorised into 3 tertiles according to the magnitude of the induction rate change among pregnancies 41[+0] weeks or more between the time periods and the relative risks (period 2 versus period 1) for outcomes were individually computed for each hospital tertile category.

[§]p-Value for linear trend of risk ratios.

[¶]BMI = $kg/m^2$; kg is a person's weight in kilograms and $m^2$ is height in metres squared.

**Nordic countries include Denmark, Finland, Iceland, Norway, and Sweden.

ARR, adjusted risk ratios; BMI, body mass index; CI, confidence interval; CS, cesarean section.

both the strata-specific RRs (period 2 versus period 1) and the overall p-value for homogeneity of the obtained RRs.

The hospitals were divided into tertiles by their quota induction rate during period 2 divided by the induction rate during period 1. The first tertile consisted of hospitals where the induction rate increased <49% (n = 16), the second tertile consisted of hospitals where the induction rate increased between 49% and 78% (n = 17), and hospitals where the induction rate increased more than 79% between period 1 and 2 (n = 15) were designated to tertile 3.

For peri/neonatal death among pregnancies $41^{+0}$ weeks or more, all RR point estimates were below unity, although not always statistically significantly so. No significant heterogeneity over the strata was detected. The RR for emergency cesarean section was above unity among nearly all strata (not of multiparous women with BMI $\geq$30) with significant heterogeneity only for parity ($p = 0.047$). The adjusted RR for peri/neonatal death (period 2 versus period 1) was about the same magnitude among infants born at $41^{+0}$ to $41^{+6}$ weeks (adjusted RR 0.53; 95% CI [0.39, 0.72]) as it was for infants born at $42^{+0}$ weeks or more (adjusted RR 0.59; 95% CI [0.25, 1.40]); $p$-value for homogeneity = 0.823. Thus, the reduced mortality could not entirely be explained by a reduced number of postterm pregnancies.

For comparison, the table also displays the RRs for peri/neonatal death, period 2 versus period 1, among children born at $39^{+0}$ to $40^{+6}$ weeks. The adjusted RR (0.86; 95% CI [0.72, 1.02]) indicates that the risk for peri/neonatal death also has decreased for children born at 39 to 40 weeks though results were not significant in this group. However, the change between the periods was significantly larger among children born at $41^{+0}$ weeks or later (adjusted RR 0.53; 95% CI [0.40, 0.71]); $p$-value for homogeneity = 0.004 (**Table 4**). The adjusted RR for cesarean section, period 2 versus period 1, was significantly higher for women delivered at 39 to 40 weeks (adjusted RR 1.14; 95% CI [1.11, 1.18]) than the corresponding adjusted RR was for women delivered at $41^{+0}$ weeks or more (adjusted RR 1.08, 95% CI [1.05, 1.11]); $p$-value for homogeneity = 0.006 (**Table 4**).

Exclusions of births during 2017 ($N$ = 23,946) or infants with significant congenital malformations ($N$ = 3,113) did not alter the adjusted RR for peri/neonatal death among pregnancies $41^{+0}$ weeks or more, adjusted RR 0.53 (95% CI [0.39, 0.72]) and 0.54 (95% CI [0.40, 0.74]), respectively (**S2 Table**).

## Discussion

In this population-based study including more than 150,000 deliveries at $41^{+0}$ gestational weeks or more, we evaluated the association between a more active management of late- and postterm pregnancies in Sweden. The rate of induction among singleton pregnancies lasting $41^{+0}$ weeks or more increased from 33.7% during period 1 (2017–2019) to 52.4% during period 2 (2020–2023) with a sharp rise between 2019 and 2020. A 47% reduced risk of peri/neonatal deaths during period 2 compared to period 1 was found, from 1.7 per thousand to 0.9 per thousand born infants in absolute values. No heterogeneities of the protective effect (period 2 versus period 1) were indicated along parity, BMI, and country of birth groups. In addition, during period 2, we observed a decreased risk of the composite adverse neonatal outcome, as demonstrated by reduced number of stillbirths, peri/neonatal deaths, and infants with birth trauma.

We consider the decline in peri/neonatal mortality and adverse neonatal outcome in pregnancies lasting 41 weeks or more to be attributed with a more active management of late-term pregnancies. Both increased interventions such as labour induction and more surveillance of late-term pregnancies may have contributed to a more favourable perinatal outcome during period 2. In support of labour induction as the main explanation of our findings, we found that hospitals belonging to the tertile with the highest induction rate had a significantly lower peri/neonatal mortality than hospitals belonging to the tertile with the lowest induction rate.

A concern is the higher rate of cesarean section, which may result from increased labour inductions. However, no such effect has been reported in most other clinical trials or observational studies [9,12,14–16,34]. The higher cesarean section rate observed herein could therefore have other explanations. A higher rate of cesarean section during the COVID-19 pandemic 2020 to 2022 than before the pandemic has been reported in some studies [35–37],

although other studies have not found any change [38,39]. A higher rate of cesarean sections was also observed for pregnancies lasting 39 to 40 gestational weeks, and the increase was significantly higher in the 39 to 40 week group than in the 41 to 42 week group. Further investigation is needed but it is likely that the increase is due to several factors [40].

Our finding of reduced perinatal mortality (adjusted RR 0.52; 95% CI [0.38, 0.69]) after changing to an active management of late- and postterm pregnancies are in line with results a systematic review [11] which have evaluated induction at 41 weeks versus expectant management and induction at 42 weeks in 3 randomised trials (N = 5,161 women) [12,15,41]. Data from the 2 largest trials [12,15] (N = 4,561 women) were included in an individual participant data meta-analysis in this review [11]. Early induction significantly reduced the composite outcome of perinatal mortality and severe neonatal morbidity and perinatal mortality alone (perinatal mortality: OR 0.21; 95% CI [0.06, 0.78]) without increasing the cesarean section rate or maternal morbidity. In addition, the study showed that early induction reduced the composite outcome more in nulliparous than in multiparous women.

A recent Finnish trial (N = 381 women) compared induction at $41^{+0}$ weeks with expectant management and labour induction at $41^{+5}$ to $42^{+1}$ weeks in nulliparous women with an unripe cervix [14]. No statistically significant differences were found in rates of composite adverse neonatal outcome, and the total operative delivery rate, including both cesarean section and instrumental delivery, was lower in the early induction group than in the expectant management group.

Individual randomised trials tend to lack statistical power for rare outcomes such as perinatal mortality. Large observational studies using real-world data can provide additional useful information to close this gap of knowledge. However, observational data have shown conflicting results. A Scottish cohort study demonstrated a substantial lower perinatal mortality among infants to women induced at 41 weeks as compared with expectant management (OR 0.30; 95% CI [0.20, 0.46]) without increasing the risk of operative delivery [34]. In Denmark, the effect of changing the national guidelines for labour induction from $42^{+0}$ to $41^{+3-5}$ weeks in 2012 has been evaluated in 2 observational studies [9,16]. While neither study found any effect on cesarean section rates, only Zizzo and colleagues [16] found a significant decline in stillbirth and perinatal mortality after implementation of the new guidelines. Another observational Dutch study compared induction of labour at $41^{+0}$ to $42^{+0}$ weeks, separated into three 2-day interval groups, to expectant management using propensity score matching [5]. In all 2-day interval groups at 41 weeks, induction reduced stillbirth compared with expectant management while rates of cesarean section were elevated for both nulliparous and multiparous women. Lindegren and colleagues found in an earlier Swedish register-based study including pregnancies lasting $41^{+3}$ weeks or more between 2001 and 2013 that among primiparas, a decreased risk of infants with low 5-min Apgar scores and meconium aspiration was found with active versus expectant management [17]. An increased risk of cesarean section was found for both nulliparous and multiparous women at units with active versus expectant management while no difference in perinatal mortality was observed.

An important aspect is women's experiences of induction. In the present study, we found no difference in childbirth experience or self-rated health after birth between the 2 periods. There is little knowledge from randomised trials on what pregnant women consider important regarding induction versus expectant management in late- and postterm pregnancies. A randomised trial from Norway showed that 74% of women preferred induction of labour at 41 weeks instead of expectant management after 41 weeks [42]. Nilver and colleagues evaluated childbirth experience in SWEPIS [15] and found no difference between women randomised to early induction at 41 weeks versus expectant management and induction at 42 weeks [43]. Overall, women's ratings of their childbirth experience were high. As part of the INDEX trial,

Keulen and colleagues evaluated the preference for management strategy at 41weeks with a questionnaire [44]. Of 782 invited women, 604 answered; 44.7% preferred induction at 41 weeks, 42.1% preferred expectant management, while 12.2% of the women did not have a preference. The vast majority had a positive birth experience. Women's preference for induction or expectant management was influenced by anxiety and quality of life problems (induction) and a wish for a natural birth (expectant management). A qualitative interview study by Lou and colleagues performed 4 to 8 weeks after an induction at late term showed that most women considered the induction to be a positive experience [45]. A systematic review of qualitative evidence highlighted the importance of individualised information and education of alternatives and details of procedures as well as risk and benefits to women with uncomplicated postterm pregnancies [46].

The strengths of this study include its large sample size allowing for evaluation of rare but important outcomes as perinatal mortality and its population-based design, with the use of individual-level data from high-quality nationwide registries with almost complete follow-up. Data on exposures and outcomes were prospectively collected. In Sweden, there is universal and free access to healthcare minimising selection bias due to differences in socioeconomic status. Furthermore, we were able to adjust risk estimates for important confounders.

Some limitations should be considered when interpreting the results. We had no data on monitoring of fetal health or other treatment details. The national guidelines [24] were issued in 2021 but in most health care regions, a change to a more proactive management was initiated already in 2019. Questions have then been aroused regarding side effects of inductions, and how an optimal outcome can be reached with minimal intervention in the normal process of labour. Further concerns have been on the effect on future pregnancies, increased workload and resource utilisation with risk of displacement consequences at the delivery wards and increased costs.

Moreover, heterogeneity of obstetric practice over the time period may have influenced findings. The significantly higher rate of hypertensive disorders and gestational diabetes seen during period 2 is most likely an effect of the change in definition of preeclampsia and implementation of new diagnostic criteria for gestational diabetes in Sweden [47–49].

Women assessed their childbirth experience by a 10-point numeric rating scale before discharge a few days after birth. This method is used nationwide in Sweden as a part of clinical routine and is an easy and valid method that correlates with other birthing experiences instruments [50]. However, we included only women giving birth at hospitals with a response rate of 80% or more resulting in a substantial proportion of missing data. Similar, there were missing values for self-rated health after delivery, and the results should therefore be interpreted with caution. Finally, as in all observational studies residual confounding by unknown or unmeasured factors may remain.

Determining the threshold for induction of late- and postterm pregnancies has been described as "the 41-to-42-week dilemma" [51]. Adopting a policy of induction at 41 weeks in Sweden has substantially increased the numbers of inductions. It is important to assess whether improved outcomes such as reduced perinatal death can be achieved with induction or increased fetal surveillance and to determine the optimal gestational threshold for induction in late-term pregnancies. The optimal gestational thresholds most probably differ according to the woman's individual risk profile, e.g., parity, previous cesarean section, and BMI, yet this could not be confirmed in the present study.

The adjusted RR of 0.52 for peri/neonatal death, period 2 versus period 1, corresponds to an extra survival of approximately 18 children per 22,000 births (the average annual number of pregnancies lasting $41^{+0}$ weeks or more in Sweden). Similarly, the adjusted RR of 1.07 for emergency cesarean section corresponds to an extra 162 cesarean sections per 22,000

deliveries. And, finally, the increased induction rate from 33.3% to 52.4% corresponds to approximately 4,000 more inductions per 22,000 deliveries at $41^{+0}$ weeks or more.

The costs and economic effects need to be taken into consideration. Routine labour induction is costly and may lead to allocation of health care resources to low risk rather than high-risk pregnancies [52,53].

Inductions at 39 to 40 weeks increased. The underlying mechanism for this finding was not investigated in this study and warrants further exploration in the future. However, results from the 35/39 trial in the UK [54] and the ARRIVE trial in the USA [7] published in 2016 and 2018, respectively, which both offered elective induction in nulliparous women at 39 weeks of gestation compared with expectant management may also have had implications for obstetric clinical practice in Sweden and other countries across the world.

Further long-term follow-up are also needed about future health and development in children born in late-and postterm pregnancies.

Shared decision-making when counselling women about induction is important. After balanced information, some women will prefer induction as an alternative at 41 weeks, others will choose expectant management to get an opportunity to enter spontaneous labour.

The main findings in this national study are that a more active management of late- and postterm pregnancies was associated with a decrease in peri/neonatal deaths and a decrease in composite adverse peri/neonatal outcome including birth trauma. Both increased interventions such as labour induction and more surveillance of late-term pregnancies may have contributed to a more favourable perinatal outcome. An increase in cesarean was observed.

The lead author (the manuscript's guarantor) affirms that this manuscript is an honest, accurate, and transparent account of the study being reported; that no important aspects of the study have been omitted; and that any discrepancies from the study as planned (and, if relevant, registered) have been explained.

## Supporting information

**S1 RECORD Checklist. Checklist of items, extended from the STROBE statement, that should be reported in observational studies using routinely collected health data.**
(DOCX)

**S1 Protocol. Study protocol as approved by the ethical committee in Sweden.**
(PDF)

**S1 Syntax Main Analysis. The syntax for the main analyses, using IBM SPSS Statistics, version 27.0.** (Armonk, NY: IBM Corp, USA).
(PDF)

**S1 Table. Results from analyses based on weighted summary of unit-specific Risk Ratios (period 2 versus period 1).**
(DOCX)

**S2 Table. Results from sensitivity analyses excluding births during 2017 or infants with significant congenital malformations.**
(DOCX)

## Author Contributions

**Conceptualization:** Karin Källén, Mikael Norman, Charlotte Elvander, Christina Bergh, Verena Sengpiel, Henrik Hagberg, Teresia Svanvik, Ulla-Britt Wennerholm.

**Data curation:** Karin Källén.

**Formal analysis:** Karin Källén.

**Funding acquisition:** Mikael Norman, Christina Bergh, Verena Sengpiel, Henrik Hagberg, Ulla-Britt Wennerholm.

**Investigation:** Karin Källén, Mikael Norman, Charlotte Elvander, Christina Bergh, Verena Sengpiel, Henrik Hagberg, Teresia Svanvik, Ulla-Britt Wennerholm.

**Methodology:** Karin Källén, Mikael Norman, Charlotte Elvander, Christina Bergh, Verena Sengpiel, Henrik Hagberg, Teresia Svanvik, Ulla-Britt Wennerholm.

**Project administration:** Karin Källén, Ulla-Britt Wennerholm.

**Writing – original draft:** Karin Källén, Ulla-Britt Wennerholm.

**Writing – review & editing:** Karin Källén, Mikael Norman, Charlotte Elvander, Christina Bergh, Verena Sengpiel, Henrik Hagberg, Teresia Svanvik, Ulla-Britt Wennerholm.

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
