## [Editor Report · Decision Letter 0]

31 May 2024

Dear Dr Wennerholm, 

Thank you for submitting your manuscript entitled "Maternal and perinatal outcomes after implementation of a more active management in late-and postterm pregnancies in Sweden: a population-based cohort study" for consideration by PLOS Medicine.

Your manuscript has now been evaluated by the PLOS Medicine editorial staff and I am writing to let you know that we would like to send your submission out for external peer review.

Please re-submit your manuscript within two working days, i.e. by Jun 04 2024 11:59PM.

Feel free to email me at pdodd@plos.org or the team at plosmedicine@plos.org if you have any queries relating to your submission.

Kind regards,

Pippa

Philippa C. Dodd, MBBS MRCP PhD

PLOS Medicine

pdodd@plos.org

---

## [Decision Letter · Decision Letter 1]

6 Sep 2024

Dear Dr Wennerholm,

Many thanks for submitting your manuscript "Maternal and perinatal outcomes after implementation of a more active management in late-and postterm pregnancies in Sweden: a population-based cohort study" (PMEDICINE-D-24-01706R1) to PLOS Medicine. The paper has been reviewed by subject experts and a statistician; their comments are included below and can also be accessed here: [LINK]

As you will see, the reviewers had mixed opinions regarding the manuscript. After discussing the paper with the editorial team and an academic editor with relevant expertise, I'm pleased to invite you to revise the paper in response to the reviewers' comments. We plan to send the revised paper to some or all of the original reviewers, and we cannot provide any guarantees at this stage regarding publication.

We ask that you submit your revision by Sep 27 2024 11:59PM. However, if this deadline is not feasible, please contact me by email, and we can discuss a suitable alternative.

Don't hesitate to contact me directly with any questions (pdodd@plos.org). 

Best regards, 

Philippa 

Philippa Dodd, MBBS MRCP PhD 

Senior Editor

PLOS Medicine

pdodd@plos.org

Comments from the reviewers: 

Reviewer #1: See attachment

Michael Dewey

Reviewer #2: Thank you for sending me this paper to review. The authors are to be congratulated on their commitment to evaluating what often goes unevaluated, policy change, and for including women's experiences in their evaluation. 

Overall however I am not sure that this manuscript adds substantially to the existing evidence regarding induction of labour. The authors report a change in practice between two time periods and then a change in outcomes. These time periods also encompass the pandemic and insufficient reference is made to this. The methods do not account sufficiently for comorbidity and the results are not clearly stated. The authors do not make sufficient reference to the background literature and context - they state that the observed increase in induction at 39 weeks needs further investigation, but do not reference the 35/39 trial (published 2016) and ARRIVE trial (published 2018) which both made the offer of induction at 39 weeks much more common across the world. I think it is a well conducted study and with revision is suitable for publication in another journal, but at present, I do not think it meets the high standards of PLOS Medicine.

Specific comments:

Abstract: should state what the policy was prior to the change in the background section, at present this could only be inferred from the results.

Introduction: 

Lines 104-107: the existing evidence does not suggest a dichotomy (as suggested here) but rather, a risk that increases with advancing gestational age.

Lines 108-111: The literature does not universally confirm this to be the case, for example, the ARRIVE trial (doi 10.1056/NEJMoa1800566).

Methods:

Line 153: When are these outcomes, of birth experience and general health, recorded?

Line 171: When was the fetus in cephalic presentation - at birth?

Primary outcomes: how were these chosen? Scalp trauma seems relatively minor (and common at instrumental birth) compared to prolonged neonatal admission. How were the Apgar score and length of stay cut offs chosen? References would be helpful.

Secondary outcomes: line 194, do you mean obstetric anal sphincter injury or full sphincter rupture (ie 3c tear)? Is level of PPH recorded?

Statistical analysis: Line 200: Why did you not adjust for other maternal comorbidities such as diabetes and hypertension, given that you had the information from ICD-10 codes?

Results: 

This section needs to be reorganised with Table 1 discussed early. I think it's important for the reader's understanding to present very early in the results, the finding that of women over 41 weeks, the proportion going past 42 dropped from 24 to 8 percent (so by 2/3) between the two periods.

In general, you need more numbers in the results - at the moment the results only say indicative sentences rather than giving numbers. E.g. 'A very prominent increase of the induction rate between the two time periods (2017 - 2019 and 2020 - 2023, respectively) was noted among births at 41+1 to 41+6 weeks' - this should have what the induction rate was and what it became in it.

Line 273 onwards about variation should be in a new paragraph.

Please include a CI with all reporting of adjusted and unadjusted rate ratios.

Lines 340 onwards: how were these thresholds arrived at? In general, I think 'tertile' is not a word that's used much - in English we tend to use thirds, I would suggest highest third of … etc. 

Line 350: please include RR here and when making statements about RRs.

Discussion:

The authors do not make sufficient reference to the background literature and context - fitting it within the very narrow question of when to put the cut off (41+6 or later) rather than within the broader understanding of when IOL for term, low risk pregnancies should be. The authors state that the observed increase in induction at 39 weeks needs further investigation, but do not reference the 35/39 trial (published 2016) and ARRIVE trial (published 2018) which both made the offer of induction at 39 weeks much more common across the world. I would strongly suggest revision with this in mind.

Reviewer #3: The authors of the article "Maternal and perinatal outcomes after implementation of a more active management in late-and post-term pregnancies in Sweden: a population-based cohort study" have conducted a significant study. They have demonstrated in a large population-based cohort study that active pregnancy management after the completion of 41 weeks of pregnancy reduces perinatal mortality, as well as neonatal mortality and morbidity. However, they also found an unexplained increase in the cesarean section rate. The research is well-designed, and the methodology is very precisely described, including the statistical analysis used. The results are accurately presented. Due to the extensiveness and additional sensitivity analyses, reading the results is tiring, but the tables and graph complement them perfectly. The discussion summarizes well the main findings of the research and compares them accordingly with other research. They also adequately identify the shortcomings of the research. The conclusions are consistent with the results.

Although there are already several articles on the subject of induction of labor before and after 41 weeks of pregnancy and there has already been some discussion on the topic at congresses, population-based real-word data research contributes significantly to decisions in clinical practice. The biggest issue, which the research does not really address, is the increase in the rate of caesarean sections. This is a crucial area that requires further investigation. In the discussion, they adequately define this problem with directions for further research. The cited literature is relevant.

I did not detect any major flaws in the article. Of the smaller ones, I would mention the title of Table 1, where a reader can understand at first glance that all are excluded:

"Singleton pregnancies, cephalic presentations, elective caesarean sections excluded."

---

* Please upload any figures associated with your paper as individual TIF or EPS files with 300dpi resolution at resubmission; please read our figure guidelines for more information on our requirements: http://journals.plos.org/plosmedicine/s/figures. While revising your submission, please upload your figure files to the PACE digital diagnostic tool, https://pacev2.apexcovantage.com/. PACE helps ensure that figures meet PLOS requirements. To use PACE, you must first register as a user. Then, login and navigate to the UPLOAD tab, where you will find detailed instructions on how to use the tool. If you encounter any issues or have any questions when using PACE, please email us at PLOSMedicine@plos.org.

* Please ensure that the study is reported according to the STROBE guideline and include the completed STROBE checklist as Supporting Information. When completing the checklist, please use section and paragraph numbers, rather than page numbers. Please add the following statement, or similar, to the Methods: "This study is reported as per STROBE guideline (S1 Checklist)."

FIGURES AND TABLES

SUPPLEMENTARY MATERIAL

REFERENCES

OBSERVATIONAL STUDIES

* Abstract: Please include the study design, population and setting, number of participants, years during which the study took place (enrollment and follow up), length of follow up, and main outcome measures.

* [FOR POPULATION HEALTH/REGISTRY STUDIES] Please ensure that the study is reported according to the RECORD guideline (available from https://www.record-statement.org) and include the completed checklist as Supporting Information. Please add the following statement, or similar, to the Methods: "This study is reported as per the Reporting of Studies Conducted using Observational Routinely-Collected Data (RECORD) guideline (S1 Checklist)." When completing the checklist, please use section and paragraph numbers, rather than page numbers.

* For all observational studies, in the manuscript text, please indicate: (1) the specific hypotheses you intended to test, (2) the analytical methods by which you planned to test them, (3) the analyses you actually performed, and (4) when reported analyses differ from those that were planned, transparent explanations for differences that affect the reliability of the study's results. If a reported analysis was performed based on an interesting but unanticipated pattern in the data, please be clear that the analysis was data driven. 

* Please state in the Methods section whether the study had a prospective protocol or analysis plan. If a prospective analysis plan (from your funding proposal, IRB or other ethics committee submission, study protocol, or other planning document written before analyzing the data) was used in designing the study, please include the relevant document(s) with your revised manuscript as a Supporting Information file to be published alongside your study and cite it in the Methods section. A legend for this file should be included at the end of your manuscript. If no such document exists, please make sure that the Methods section transparently describes when analyses were planned, and when/why any data-driven changes to analyses took place. Changes in the analysis, including those made in response to peer review comments, should be identified as such in the Methods section of the paper, with rationale.

---

## [Decision Letter · Decision Letter 2]

5 Nov 2024

Dear Dr. Wennerholm,

Thank you very much for re-submitting your manuscript "Maternal and perinatal outcomes after implementation of a more active management in late-and postterm pregnancies in Sweden: a population-based cohort study" (PMEDICINE-D-24-01706R2) for review by PLOS Medicine.

Thank you for your detailed response to the editors' and reviewers' comments. I have discussed the paper with my colleagues and the academic editor, and it has also been seen again by two of the original reviewers. The changes made to the paper were mostly satisfactory to the reviewers. As such, we intend to accept the paper for publication, pending your attention to the reviewers' and editors' comments below in a further revision. When submitting your revised paper, please once again include a detailed point-by-point response to the editorial comments.

[LINK]

In revising the manuscript for further consideration here, please ensure you address the specific points made by each reviewer and the editors. In your rebuttal letter you should indicate your response to the reviewers' and editors' comments and the changes you have made in the manuscript. Please submit a clean version of the paper as the main article file. A version with changes marked must also be uploaded as a marked up manuscript file. Please also check the guidelines for revised papers at http://journals.plos.org/plosmedicine/s/revising-your-manuscript for any that apply to your paper.

We ask that you submit your revision within 1 week (Nov 12 2024). However, if this deadline is not feasible, please contact us by email, and we can discuss a suitable alternative.

Please do not hesitate to contact us directly with any questions (pdodd@plos.org or atosun@plos.org). If you reply directly to this message, please be sure to 'Reply All' so your message comes directly to our inbox.

We look forward to receiving the revised manuscript.   

Sincerely,

Alexandra Tosun, PhD

Associate Editor

[on behalf]

Philippa Dodd, MBBS MRCP PhD

Senior Editor 

PLOS Medicine

plosmedicine.org

Comments from Reviewers:

Reviewer #1: The authors have addressed my points.

Just to clear up a small misunderstanding about titles not in English. I know they do not have an official translation but the authors could supply their version or a comment so that people know what the content might be to save them going to the trouble of downloading it, putting it through a machine translation and then finding out it was irrelevant to their interests.

Michael Dewey

Reviewer #2: Responses to my queries are adequate, thank you.

[LINK]

Requests from Editors:

CODE AVAILABILITY

Thank you for providing a document with the syntax for the main analyses (S4 Appendix). We kindly ask you to include a sentence about the availability of the code in the data availability statement in the online submission form.

ABSTRACT

1) l.72: Please include the important dependent variables that are adjusted for in the analyses.

2) ll.74-75: When reporting means, please provide the standard deviation.

3) l.76: Please specify what the number 1000 refers to, at least the first time you report it.

4) We suggest reporting statistical information as follows to improve clarity for the reader "22% (95% CI [13%,28%]; p</=)". When reporting 95% CIs please separate upper and lower bounds with commas instead of hyphens as the latter can be confused with reporting of negative values. For example, lines 76-77: “(adjusted RR 0.52; 95% CI [0.38,0.69]; p<0.001)”

5) In the last sentence of the Abstract Methods and Findings section, please describe the main limitation(s) of the study's methodology.

6) If you agree, we suggest adding a sentence about the implications of the study in the Abstract Conclusion, as is done in the Author Summary under 'What do these findings mean?’.

AUTHOR SUMMARY

Thank you for providing the Author Summary. Please revise with regard to formatting. Each sub-heading (please note the question format) should contain 2-3 single sentence, concise bullet points containing the most salient points from your study. In the final bullet point of 'What Do These Findings Mean?', please include the main limitations of the study in non-technical language. 

INTRODUCTION

1) l.107: Please note that ‘(40+6 weeks)’ should have two right parentheses.

2) l.129: Please define ‘WHO’ at first use.

METHODS AND RESULTS

1) Table 1: Please define ‘Nordic country’ below the table. 

2) Table 1: For ‘child characteristics’, we suggest changing the first one to ‘Sex, Male’. Please note that the terms gender and sex are not interchangeable (as discussed in https://www.who.int/health-topics/gender#tab=tab_1 ); please use the appropriate term.

3) Table 2: Please define ‘CI’ below the table. Please see the comments regarding reporting of statistical information (under Abstract) and revise accordingly.

4) Table 3: Please define ‘CI’ and ‘RR’ below the table.

5) Table 4: Please define ‘CI’ and ‘RR’ below the table. Please add a unit for BMI. Please define ‘Nordic country’ below the table. Please add a brief explanation about the hospital induction change tertile as tables should be self-explanatory on a stand-alone basis.

6) Figure 1: Please define ‘CS’. Please note that words in the image are underlined in red, probably by the spell check function.

7) ll.388-391: Apologies if I missed it, but are these results presented anywhere in a supplementary table? Please refer to the relevant table or provide such a table.

DISCUSSION

1) Please remove any subheadings from the discussion section, including the ‘conclusion’ subheading.

2) l.400: “per thousand”? – please specify.

SUPPLEMENTARY MATERIAL

In the published article, supporting information files are accessed only through a hyperlink attached to the captions. For this reason, you must list captions at the end of your manuscript file. You may include a caption within the supporting information file itself, as long as that caption is also provided in the manuscript file. Do not submit a separate caption file.

When SI files are contained with a single file:

Please label the file as ‘S1 Supporting Information’.

Please apply alphabetical labelling to each table and figure contained within the S1 file. For example, ‘Fig A’ to ‘Fig Z’ and ‘Table A’ to ‘Table Z’.

Plain text does not need to be labelled and can just be given a title as necessary. For example, ‘Statistical Analysis Plan’.

Please cite tables/figures as ‘Fig A in S1 Supporting Information’ and/or ‘Table A in S1 Supporting Information’, for example.

Please cite plain text as, ‘Statistical Analysis Plan in S1 Supporting Information’, for example.

When SI files are uploaded as separate files:

Please label tables as ‘S1 Table’ (so on) and figures as ‘S1 Fig’ (and so on).

Any additional documents (protocols/analysis plans etc.) can be labelled as ‘S1 Protocol’, for example. Please cite items as exactly as labelled.

SOCIAL MEDIA

To help us extend the reach of your research, please provide any X (formerly known as Twitter) handle(s) that would be appropriate to tag, including your own, your co-authors’, your institution, funder, or lab. Please enter in the submission form any handles you wish to be included when we post about this paper.

General Editorial Requests

---

## [Editor Report · Decision Letter 3]

18 Nov 2024

Dear Dr Wennerholm, 

On behalf of my colleagues and the Academic Editor, Gordon C Smith, I am pleased to inform you that we have agreed to publish your manuscript "Maternal and perinatal outcomes after implementation of a more active management in late-and postterm pregnancies in Sweden: a population-based cohort study" (PMEDICINE-D-24-01706R3) in PLOS Medicine.

I appreciate your thorough responses to the reviewers' and editors' comments throughout the editorial process. We look forward to publishing your manuscript, and editorially there are only a few remaining minor stylistic/presentation points that should be addressed prior to publication. We will carefully check whether the changes have been made. If you have any questions or concerns regarding these final requests, please feel free to contact me at atosun@plos.org.

Please see below the minor points that we request you respond to:

1) Author Summary: Please add questions marks to the three headings (i.e. Why was this study done?, What did the researchers do and find?, What do these findings mean?)

2) l.319 (track changes version): Please define ‘CI’ at first use.

3) Please clarify whether the section "Dissemination to participants and related patient and public communities" should be included in the Methods section of your manuscript. If so, please move it to the Methods section. If not, we suggest removing it from the end of the main text.

Before your manuscript can be formally accepted you will need to complete some formatting changes, which you will receive in a follow up email (including the editorial points above). Please be aware that it may take several days for you to receive this email; during this time no action is required by you. Once you have received these formatting requests, please note that your manuscript will not be scheduled for publication until you have made the required changes.

PRESS

Sincerely, 

Alexandra Tosun, PhD

Associate Editor

[on behalf of]

Philippa C. Dodd, MBBS MRCP PhD 

Senior Editor 

PLOS Medicine